# Tunable Donor–Acceptor Linear Conjugated Polymers Involving Cyanostyrylthiophene Linkages for Visible-Light-Driven Hydrogen Production

**DOI:** 10.3390/molecules28052203

**Published:** 2023-02-27

**Authors:** Dongnai Ye, Lei Liu, Yujie Zhang, Jiabin Qiu, Zhirong Tan, Yuqin Xing, Shiyong Liu

**Affiliations:** 1Jiangxi Provincial Key Laboratory of Functional Molecular Materials Chemistry, College of Materials, Metallurgical and Chemistry, Jiangxi University of Science and Technology, Ganzhou 341000, China; 2School of Chemistry and Chemical Engineering, Gannan Normal University, Ganzhou 341000, China

**Keywords:** direct C-H arylation polymerization, linear donor–acceptor conjugated polymers, photocatalysis, visible-light-driven hydrogen evolution

## Abstract

In this paper, an atom- and step-economic direct C-H arylation polymerization (DArP) strategy was developed to access cyanostyrylthiophene (CST)-based donor–acceptor (D–A) conjugated polymers (CPs) used for photocatalytic hydrogen production (PHP) from water reduction. The new CST-based CPs **CP1**–**CP5** with varied building blocks were systematically studied by X-ray single-crystal analysis, FTIR, scanning electron microscopy, UV-vis, photoluminescence, transient photocurrent response, cyclic voltammetry measurements, and a PHP test, which showed that the phenyl-cyanostyrylthiophene-based **CP3** exhibits a superior hydrogen evolution rate (7.60 mmol h^−1^ g^−1^) compared to other conjugated polymers. The structure–property–performance correlation results obtained in this study will provide an important guideline for the rational design of high-performance D–A CPs for PHP applications.

## 1. Introduction

Energy consumption is ever increasing with the rapid development of human society. The exploration of clean and renewable energies has become the top priority of governments all over the world. Hydrogen energy is regarded as one of the most promising alternatives to traditional fossil fuels. Solar-driven hydrogen production from water, i.e., photocatalytic hydrogen production (PHP), is a potential way to solve environmental issues and the global energy crisis [1,2,3,4,5,6,7].

In recent years, π-conjugated polymers (CPs), consisting of alternating double and single bonds with C, H, N, O, and S light elements, have emerged as a type of promising soft photocatalyst due to their high stability and their strong absorptance toward visible light [8,9,10,11,12,13,14,15]. In comparison to state-of-the-art polymeric graphitic carbon nitride (g-C_3_N_4_) with relatively fixed structures [10,16], CPs with alternating electron donors’ (Ds’) and acceptors’ (As’) building blocks, i.e., D–A CPs, are expected to be potentially efficient photocatalysts, due to the abundance of D–A interactions, which can finely modulate the opto-electronic properties by varying the D/A ratios or by adjusting π-conjugated structures [14,17,18,19,20]. The highest occupied molecular orbital (HOMO) and the lowest unoccupied molecular orbital (LUMO), corresponding to the valence and conduction bands of an inorganic semiconductor, are governed, respectively, by the electron-donating and electron-accepting building blocks, which endows D–A CPs with a wide range of tunabilities regarding light absorption, frontier molecular orbital (FMO) levels, and photocatalytic redox capacity [14,18,21].

Diarylethenes are types of significant building blocks of the polymer semiconductors with high charge mobility for organic electronic devices, e.g., poly(thienylenevinylenes) (PTVs) [22,23]. The cyanostyrylthiophenes (CSTs), as diarylethene derivatives involving D–A architecture, are expected to serve as excellent platforms for efficient D–A CP-based photocatalyst in PHP reactions. Meanwhile, the CST moieties also have the distinct advantages of facile synthesis via Knoevenagel condensation and tunable functionalization at varied positions of the vinylene linker. With these in mind, we designed and synthesized a series of CST-based CPs (Figure 1) in an atom- and step-economic way via direct C–H arylation polymerization (DArP) from (*E*)-3-(6-bromopyridin-3-yl)-2-(thiophen-2-yl) acrylonitrile (**CST-BPD**), (*E*)-3-(5-bromothiophen-2-yl)-2-(thiophen-2-yl)acrylonitrile (**CST-BT**), (*E*)-3-(4-bromophenyl)-2-(thiophen-2-yl)acrylonitrile (**CST-BP**), (*E*)-3-(4-bromonaphthalen-1-yl)-2-(thiophen-2-yl)acrylonitrile (**CST-BN**), and (*E*)-3-(4-bromo-3-methylphenyl)-2-(thiophen-2-yl)acrylonitrile (**CST-BMP**). The effects of the electron-donating capabilities and molecular geometries of these CST-based building blocks on opto-electronic properties and catalytic performances were systematically investigated by varied characterizations and PHP reactions. The corresponding CPs—**CP1**, **CP2**, **CP3**, **CP4** and **CP5**—were studied for visible-light-driven PHP reactions from water. Among these CPs, **CP3** showed the highest PHP activity with a hydrogen evolution rate (HER) of 7.60 mmol h^−1^ g^−1^ that was achieved when water and ascorbic acid (AA) were used as proton source and sacrificial-electron donor (SED), respectively. The results of the PHP performance indicate that the steric hindrance of CST building blocks have a dominant effect on PHP performance, which provides a desirable guidance for the rational design of D–A type CP-based photocatalysts.

## 2. Results and Discussion

### 2.1. Synthesis and Characterization of the CPs

As shown in Figure 1, a series of CST-based polymers, **CP1**–**CP5**, were synthesized separately in five pots of Pd-catalyzed C–H direct arylated coupling polymerization [24,25,26] starting from building blocks **CST-BPD**, **CST-BT**, **CST-BP**, **CST-BN**, and **CST-BMP**, which were facilely obtained via Knoevenagel condensation between varied bromo-substituted aromatic aldehydes and 2-(thiophen-2-yl)acetonitrile (Th-ACN). The structures of these CST-based CPs were characterized by Fourier transform infrared (FT-IR) spectra (Figure 2). All CPs displaying peaks at ~1640 cm^−1^ wavelength were assigned to the stretching vibration of C=C double bonds, revealing the presence of vinylene linkers. The infra-red absorption peaks at about 2232 cm^−1^ were the skeleton vibration of the –CN groups in all these CPs.

To understand the structural correlation between the monomers and the corresponding polymers, the optimal geometries and the dihedral angles of the CST-based monomers and dimers were predicted by a density-function-theory (DFT) calculation. When considering the coupled unit as a system, the dihedral angles between the donor units of **CST-BPD**, **CST-BT**, **CST-BP**, **CST-BN**, and **CST-BMP** and the thiophenyl unit were 1.2°, 1.5°, 30.1°, 52.4°, and 53.6°, respectively (Appendix A). Due to the steric effect, a large dihedral angle was observed between the two units. The dihedral angles of pyridine and thiophene rings were similar, which was helpful in revealing the effect of the electron-donating ability of the pyridine and thiophene units on the opto-electronic properties and performance. The planarity of **CP3** showed little change owing to the equivalent thiophene-thiophene rings involved. The steric hindrance of the naphthalene ring and the methyl group enlarged the dihedral angles of **CP4** and **CP5**, which would be less conducive to the charge transfer.

In the CST-based monomers, the dihedral angles between the donor and acceptor units of **CST-BPD**, **CST-BT**, **CST-BP**, **CST-BN**, and **CST-BMP** were 28.0°, 26.0°, 23.9°, 46.4°, and 15.1°, respectively (Appendix A). This was because vinyl can form a strong conjugation effect with the donor and acceptor units; thus, the dihedral angles between them were greatly changed. In the dimer models of **CST-BPD**, **CST-BT**, **CST-BP**, **CST-BN**, and **CST-BMP**, the dihedral angles between the donor unit and the acceptor unit were changed to 29.0°, 28.1°, 20.8°, 43.1°, and 19.9°, respectively (Appendix A). Therefore, this trend of change can help us predict whether the dihedral angle of polymer molecules is favorable for photocatalytic hydrogen production. For example, the dihedral angle of **CP5** increased dramatically, which would be detrimental to the transport of photogenerated carriers.

The above point was also experimentally supported by single-crystal X-ray analysis. To explore the impact of CST substitutes, the single crystals of monomers **CST-BT**, **CST-BP**, and **CST-BMP** were successfully grown and subjected to X-ray crystalline structural analysis (Figure 3). Among the monomers, CST-BT and CST-BMP were two newly reported crystals (CCDC2190738 and CCDC2190742, CIF files in SI). The results demonstrated that the molecules of **CST-BT**, **CST-BP**, and **CST-BMP** pertained to the triclinic space group. Z = 4, with cell parameters of a = 6.8270 Å, b = 23.045 Å, c = 7.4519 Å, α = 90°, β = 97.604°, and γ = 90° for **CST-BT**; Z = 4, with cell parameters of a = 3.9619 Å, b = 24.465 Å, c = 12.6409 Å, α = 90°, β = 96.409°, and γ = 90° for **CST-BP**. Z = 2, with cell parameters of a = 7.5030 Å, b = 8.5195 Å, c = 11.1655 Å, α = 86.355°, β = 79.693°, and γ = 67.892° for **CST-BMP**. For the one-dimensional chain structures, the intermolecular hydrogen bonds were formed in **CST-BT**, **CST-BP**, and **CST-BMP** with distances of 2.496 Å, 2.638 Å, and 2.718 Å, respectively. Based on the data, the spatial potential resistance effect had a significant impact on the distance of the CST unit.

To gain deeper insight into the morphologies of varied CST-based CPs at micro- and nanoscales, all the CPs were investigated by scanning electron microscopy (SEM). The morphology and structure of photocatalysts in promoting the separation of photogenerated carriers contribute to the improvement of photocatalytic activity. **CP1**–**CP5** each displayed a well-defined dimension and morphology (Figure 4), which showed that the morphologies of **CP1**–**CP5** are different from each other’s. The enhanced planarities of starting monomers **CST-BPD** and **CST-BT** endowed **CP1** and **CP2** a tightly packed irregular morphology. Nevertheless, **CP4** and **CP5** displayed a dendritic-like morphology, which was consistent with the spatial steric resistance of the naphthalene ring and the methyl group. Compared with the other CPs, the SEM morphologies of **CP3** had a more regular lamellar compact structure, which was instrumental in charge transport and separation of carriers to the photocatalytic reaction.

### 2.2. Opto-Electronic Properties of the CPs

The opto-electronic properties of **CP1**–**CP5** were systematically studied by UV-vis absorption, photoluminescence (PL), transient photocurrent response (TPR) and cyclic voltammetry (CV) measurements (Figure 5). The UV-vis diffuse reflectance spectroscopy (DRS) revealed that all of the CPs had extensive absorption in the range of 300–600 nm (Figure 5a), which benefits visible-light harvesting. This suggests that introducing the electron-D/A-capability units into the conjugate backbone can tune the energy level of the CPs and lead to changes in the spectral absorption of the materials alteration. The absorption trend was consistent with the colors of CPs: better absorbance toward longer wavelengths had deeper colors (insets in Figure 5a). The nitrogen atom of the pyridine ring in **CP1** had the induction effect of electron absorption and the conjugation effect of electron absorption, which led to the wide absorption range and darker color of **CP1** in the state of solid aggregation. **CP2** with a range absorption too wide led to a narrow band gap, which resulted in an easier return of electrons from the excited state to the ground state; thus, the recombination of photogenerated electron holes sharply increased, reducing the effect of hydrogen production. The optical bandgaps (*E*_g_) of **CP1**–**CP5** estimated by the plotting curves of (*αhν*)^2^ vs. *hν* were 1.88, 1.85, 1.98, 2.06, and 1.97 eV, respectively (Appendix A). Although **CP1**–**CP5** had no solubility in common solvents such as CH_2_Cl_2_, CHCl_3_ and toluene, they all could form colloidal dispersions in N-methyl-pyrrolidinone (NMP). The UV-vis spectra of **CP1**–**CP5** colloidal dispersions in NMP exhibited maximum absorption peaks at 420.5, 526.5, 455, 426.5, and 531.5 nm wavelengths, respectively (Appendix A).

Minimizing the exciton-binding energy can increase the yield of charge-carrier generation and improve photocatalytic activity [17]. Typically, the recombination of photogenerated electron-hole pairs can be evaluated by steady-state PL spectroscopy. The PL spectra of **CP1**–**CP5** were collected, as shown in Figure 5b, to facilitate a full comparison of their PL intensities, i.e., photo–to–photo conversions. **CP4** and **CP5** stimulated by a 365 nm wavelength displayed a strong fluorescence. The weaker the PL intensity, the lower the recombination rate of electron-hole (e^−^-h^+^) pairs [14], which favors to the photogenerated carrier separation. **CP1**–**CP3** had a lower PL intensity, suggesting a decreased ratio of e^−^-h^+^ recombination (insets in Figure 5b). The above results reveal that the alternating-electron D and A building blocks involved in the polymeric backbones can effectively promote the internal electron transfer and reduce the exciton recombination.

For an excellent photocatalyst, a suitable band gap is needed to ensure a sufficient light absorptance to produce e^−^-h^+^ pairs. In addition, the photogenerated electron and hole separately transfer to the photocatalytic redox reaction before being recombined. Typically, the band gap should be greater than 1.8 eV and less than 2.2 eV to simultaneously ensure the sufficient driven force for PHP and the efficient absorption toward visible light [8]. In order to obtain materials with efficient carrier separation, we hoped to minimize the overlap between the HOMO levels while also achieving the LUMO levels, to realize the spatial separation of electron-hole pairs. Here, the electrochemical property and the FMOs of **CP1**–**CP5** were estimated by CV measurements (Appendix A). The LUMO levels of **CP1**–**CP5** were calculated, *E*_LUMO_ = −4.80 − (*E*_red_ − *E*_Fc/Fc+_), and the HOMO levels were calculated based on *E*_g_ and LUMOs: *E*_HOMO_ = *E*_LUMO_ − *E*_g_. Typically, the more negative the value of LUMO is, the higher the reduction ability will be. The results showed that the LUMO levels of **CP1**–**CP5** had sufficient driving force to reduce water to obtain hydrogen. Compared with **CP1**, **CP2**, **CP4** and **CP5**, the FMOs of **CP3** weree optimal, as its LUMO orbitals made it easier to obtain electrons and its HOMO orbitals made it easier to lose its electrons. 

TPR measurement was employed to investigate the photo-to-current conversion efficiencies of **CP1**–**CP5**. As shown in Figure 5d, **CP1**–**CP3** showed the maximum photocurrent, which meant that the electron-hole separation efficiency of the **CP1**–**CP3** composite was effectively improved. Furthermore, the TPR current of **CP3** was higher than those of **CP1** and **CP2**, which also reflected its highest photocatalytic activity.

### 2.3. Photocatalytic Hydrogen Production Performances of the CPs

The PHP performances of the polymers (Figure 6) were evaluated by using water and ascorbic acid as a hydrogen source and an SED, respectively. The HERs of **CP1**–**CP5** were 0.00, 1.89, 7.60, 1.17, and 2.47 mmol h^−1^ g^−1^, respectively. The zero HER of **CP1** might be ascribed to its being ascribed to its deepest LUMO levels with the smallest driving force for the proton reduction (Figure 5c). It shows that a stronger electron-donor effect is more conducive to affecting the efficiency of hydrogen production. An HER of 7.60 mmol h^−1^ g^−1^ for **CP3** was achieved, which was much higher than that of other CPs and outperformed most of the reported linear CPs (Appendix A). Because the electron-donating ability of thiophene and thiophene is equivalent, it was hard to form a D–A effect for **CP2** that could be corroborated by the LUMO level and the HOMO level (Figure 5c). As revealed by DFT, the major factor on the PHP performance of **CP4** and **CP5** was the steric hindrance of the building blocks, which was greatly increased after introducing the naphthyl and methyl groups into **CP4** and **CP5**, respectively. The high steric hindrance decreased the planarity and the effective conjugation lengths of the polymeric chains, which was unfavorable to the intramolecular D–A interaction and the internal spatial charge carrier separation; thus, it was less conducive to the surface photocatalytic redox reaction.

Catalytic cycling stability is an important index to evaluate the practical application of photocatalysts. Here, the cycling test of the highest-performing **CP3** was evaluated and it showed no obvious decreasing trend in the continuous photocatalytic reaction (Figure 6c). After four times cycles, about 80% (0.116 mmol H_2_ evolution) of **CP3** initial H_2_ evolution was still retained. This showed that **CP3** has an excellent cycling stability toward a sacrificial PHP reaction.

## 3. Materials and Methods

### 3.1. Materials and Methods

All of the starting materials and reagents were purchased from commercial suppliers and used directly without further purification.

The NMR spectra were carried out on the Bruker Advance III 400 model 400 MHz NMR spectrometer (Germen). The single crystals structural data of **CST-BT**, **CST-BP**, and **CST-BMP** were collected by the Bruker SMART APEX II CCD instrument diffractometer (Germen) at 293 K with Mo Kα radiation upon graphite mono-chromization (λ = 0.71073). To solve the structures of **CST-BT**, **CST-BP**, and **CST-BMP**, the Siemens SHELXTL version 5 crystallographic software package [27] was used. The full-matrix least-squares techniques were used to refine. FT-IR spectra in the range of 4000–500 cm^−1^ were performed on an FT-IR spectrometer (Bruker, ALPHA). The morphology of the photocatalysts were observed by the SEM (MLA650F, American). The DRS spectra were studied using the UV-2600 scanning UV-vis spectrophotometer. The HORIBA Instruments FL-1000 fluorescence spectrometer was used to characterize PL spectra. The CV measurement was carried out on the CHI660E (Chenhua, Shanghai) electrochemical workstation, which used the normal three-electrode-cell system. In the system, glassy carbon electrode was the working electrode, Ag/AgCl electrode was the reference electrode, and platinum wire was the counter electrode. The tetra-n-butylammonium hexafluorophosphate (TBAPF_6_, 1.5 g) was dissolved by the 5 mL acetonitrile as a supporting electrolyte. The electrochemical workstation (CHI650E/700E, Shanghai) was equipped with a conventional three electrode system, which was used to characterize the TPR. In the electrochemical workstation, the configuration Ag/AgCl (saturated with KCl) was the reference electrode and the platinum plate was the counter electrode. A Gaussian 09 program using M062X-D3 density function with the 6-31+G(d,p) basis set was used for the all theoretical calculations [28,29,30]. By frequency analysis at 298 K, the chemical structure was optimized and characterized.

### 3.2. Synthesis of **CST-BPD**, **CST-BT**, **CST-BP**, **CST-BN**, and **CST-BMP**

Monomers **CST-BPD**, **CST-BT**, **CST-BP**, **CST-BN**, and **CST-BMP** were synthesized by the same procedure via Knoevenagel condensation [31,32,33,34]. The chemical structures and properties of all the monomers were characterized by ^1^H & ^13^C NMR (copies of spectra can be found in the Appendix A).

(*E*)-3-(6-bromopyridin-3-yl)-2-(thiophen-2-yl) acrylonitrile (**CST-BPD**): To a round-bottom flask containing a solution of sodium tert-butoxide (24 mg, 0.25 mmol) in 20 mL of methanol were added thiophene-2-carbaldehyde (615 mg, 5 mmol) and 6-bromonicotinaldehyde (930 mg, 5 mmol). After the reaction mixture was stirred at room temperature for 24 h, the resulting solid was filtered and further purified by recrystallization from water/methanol to yield the title compound as a green solid (1330.9 mg, 91.4%). ^1^H NMR (400 MHz, CDCl_3_) δ: 8.56 (d, *J* = 5.6 Hz, 1H), 8.28(d, d, *J* = 6.0 Hz, 1H), 7.59 (d, *J* = 8.4 Hz, 1H), 7.44 (m, 1H), 7.38 (m, 1H), 7.29 (m, 1H), 7.10(d, d, *J* = 1.6 Hz, 1H). ^13^C NMR (100 MHz, CDCl_3_) δ: 151.2, 143.4, 138.2, 136.4, 133.6, 128.7, 128.5, 128.4, 128.4, 127.5, 116.1, 109.2. 0.5 mmol.

(*E*)-3-(5-bromothiophen-2-yl)-2-(thiophen-2-yl) acrylonitrile (**CST-BT**): To a round-bottom flask containing a solution of Piperidine (0.5 mL) in 5 mL of ethanol were added thiophene-2-carbaldehyde (246 mg, 2 mmol) and 5-bromothiophene-2-carbaldehyde (382 mg, 2 mmol). The reaction temperature was 60 °C, the reaction time was 24 h, and the stirring rate was 1000 rpm. After the reaction finished, an appropriate amount of hydrochloric acid was added and the reaction mixture was washed with deionized water. The crude residue was dried with Na_2_SO_4_ and purified by column chromatography on silica gel using dichloromethane and petroleum ether as an eluent to obtain a red solid with the yield of 63.1% (373.5 mg).^1^H NMR (400 MHz, CDCl_3_) δ: 7.34 (s, 1H), 7.32 (d, *J* = 3.2 Hz, 1H), 7.30 (d, *J* = 4.8 Hz, 1H), 7.27 (s, 1H), 7.08 (d, *J* = 4.0 Hz, 1H), 7.04~7.06 (d, d, *J* = 1.2 Hz, 1H). ^13^C NMR (100 MHz, CDCl_3_) δ: 139.1, 138.3, 132.7, 131.2, 130.7, 128.3, 127.3, 126.3, 118.1, 116.9, 103.4.

(*E*)-3-(4-bromophenyl)-2-(thiophen-2-yl) acrylonitrile (**CST-BP**): To a round-bottom flask containing a solution of sodium tert-butoxide (39 mg, 0.41 mmol) in 40 mL of methanol were added 2-(thiophen-2-yl) acetonitrile (1.5 g, 8.12 mmol) and 4-bromobenzaldehyde (1.55 g, 8.12 mmol). After the reaction mixture was stirred at room temperature for 24 h, the resulting solid was filtered and further purified by recrystallization from water/methanol to yield the title compound as a yellow needle (2.14 g, 90.7%). ^1^H NMR (400 MHz, CDCl_3_) δ: 7.70 (d, *J* = 7.7 Hz, 2H), 7.58 (d, *J* =7.7 Hz, 2H), 7.39 (d, *J* = 4.2 Hz, 1H), 7.32 (d, *J* = 3.5 Hz, 1H), 7.29 (s, 1H), 7.06~7.09 (m, 1H). ^13^C NMR (100 MHz, CDCl_3_) δ: 138.9, 138.0, 132.3, 132.2, 130.5, 128.3, 127.7, 116.6, 106.8.

(*E*)-3-(4-bromonaphthalen-1-yl)-2-(thiophen-2-yl) acrylonitrile (**CST-BN**): To a round-bottom flask containing a solution of sodium tert-butoxide (9.6 mg, 0.1 mmol) in 5 mL of methanol were added thiophene-2-carbaldehyde (246 mg, 2 mmol) and 4-bromo-1-naphthaldehyde (470 mg, 2 mmol). After the reaction mixture was refluxed for 24 h, the resulting solid was filtered and further purified by recrystallization from water/methanol to yield the title compound as a green solid (589.4 mg, 86.7%). ^1^H NMR (400 MHz, CDCl_3_) δ: 8.34 (m, 1H), 8.02 (s, 1H), 7.96 (d, *J* = 7.6 Hz, 1H), 7.87 (m, 2H), 7.65 (m, 2H), 7.47 (d, d, *J* = 3.6 Hz, 1H), 7.37 (d, d, *J* = 5.2 Hz, 1H), 7.12 (d, d, *J* = 1.6 Hz, 1H). ^13^C NMR (100 MHz, CDCl_3_) δ: 137.7, 136.8, 132.5, 132.1, 130.7, 129.8, 128.3, 128.2, 128.0, 127.9, 127.8, 127.1, 126.9, 125.8, 123.9, 116.4, 110.2, 100.2.

(*E*)-3-(4-bromo-3-methylphenyl)-2-(thiophen-2-yl) acrylonitrile (**CST-BMP**): To a round-bottom flask containing a solution of sodium tert-butoxide (9.6 mg, 0.1 mmol) in 5 mL of methanol were added thiophene-2-carbaldehyde (246 mg, 2 mmol) and 4-bromo-3-methylbenzaldehyde (398 mg, 2 mmol). After the reaction mixture was refluxed for 24 h, the resulting solid was filtered and further purified by recrystallization from water/methanol to yield the title compound as a green solid (549.4 mg, 90.3%). ^1^H NMR (400 MHz, CDCl_3_) δ: 7.65 (s, 1H), 7.58 (d, *J* = 8.4 Hz, 1H), 7.56 (s, 1H), 7.38 (s, 1H), 7.32 (s, 1H), 7.26 (s, 1H), 7.07 (d, d, *J* = 1.6 Hz 1H), 2.44 (s, 3H). ^13^C NMR (100 MHz, CDCl_3_) δ: 139.0, 138.8, 138.3, 133.0, 132.5, 131.3, 128.2, 127.6, 127.5, 127.4, 126.7, 116.7, 106.5, 23.0.

### 3.3. Synthesis of **CP1**–**CP5**

Anhydrous toluene was pretreated by calcium hydride (CaH_2_) and freshly distilled.

The synthetic procedures of **CP1**–**CP5** were as follows. Pd(OAc)_2_ (5 mol%), P(t-Bu)_2_Me-HBF_4_ (10 mol%), K_2_CO_3_ (2.5 equiv.), PivOH (30 mol%) were added into Schlenk tubes, weighing substances of **CST-BPD** 100 mg for **CP1** [35]; Pd_2_(dba)_3_ (1.5 mol%), P(o-MeOPh)_3_ (3 mol%), anhydrous Cs_2_CO_3_ (2 equiv.), PivOH (30 mol%) were added into Schlenk tubes, weighing substances of **CST-BT**, **CST-BP**, **CST-BN**, **CST-BMP** 100 mg in proper order for **CP2**–**CP5** [14,24]. The mixture in the tubes was purged by repetitions of vacuuming and argon-filling (×3). Then, 5 mL of anhydrous toluene was added in the tubes. The mixture was put through freeze-vacuum-thaw cycles two times to remove dissolved air. Then, under argon atmosphere, the mixture was rigorously stirred at 150 °C for 60 h to **CP1**; and110 °C for 48 h to **CP2**–**CP5**.

After cooling to room temperature, the reaction mixture was washed by CH_2_Cl_2_ and filtered to remove the solvent. The undissolved crude CPs product on the filter paper was washed successively by CH_2_Cl_2_, methanol, water to remove the unreacted small molecules and the inorganic salt, respectively. Then the product was dried in a vacuum for 24 h at 70 °C and obtained the powder of polymers **CP1**, **CP2**, **CP3**, **CP4**, and **CP5** in yields of 23.3%, 93.4%, 77.1%, 93.6%, and 94.2%, respectively.

### 3.4. PHP Tests

A gas chromatograph (GC9790, FuLi) was linked with the photocatalytic online analysis system (LabSolar-III AG, Beijing Perfect Light) for the typical PHP test; it was equipped with the thermal conductive detector (TCD) and used argon as the carrier gas. A mixed aqueous solution containing 30 mL H_2_O and 5 g AA as the sacrificial agent was prepared for the photocatalysts **CP1**–**CP5** (10 mg) ultrasonic dispersal. The KOH solution was added to adjust the pH to 4.0. The oil pump was used to remove the dissolved air in the mixture and maintai in vacuum. To irradiate the reaction vessel, a 300 W Xe lamp (Beijing Perfect Light, PLS-SXE300) under full-arc light irradiation was used. To fix the reaction temperature at 25 °C, a flow of cooling water was used.

## 4. Conclusions

A series of tunable linear donor–acceptor conjugated-polymer photocatalysts with CST linkages were designed and facilely synthesized via an atom- and step-economic direct C–H arylation polymerization with the merits of low-cost starting material and high synthetic yields. The opto-electronic properties of **CP1**–**CP5** and their PHP reaction driven by visible light were systematically studied according to several instructive structure–property–performance correlations. (1) With the enhancement of electron-donating ability along the linear direction, CST-based conjugated polymers can have better photocatalytic performance in hydrogen evolution; (2) The higher the steric hindrance of the donor substituents, the lower the PHP performance of the CST-based CPs; (3) The introduction of phenyl into the CST-based CPs can not only broaden the light-absorption range, but also further improve the photocatalytic reduction ability, which is conducive to improving PHP performance. As a result, **CP3** exhibits the highest HER among the five linear CP-based photocatalysts and outperforms most of the reported linear CPs. Such π-conjugated materials with a definite structure offer a new prototype for further developing high-efficiency photocatalysts toward solar to chemical energy conversion.

## Figures and Tables

**Figure 1 molecules-28-02203-f001:**
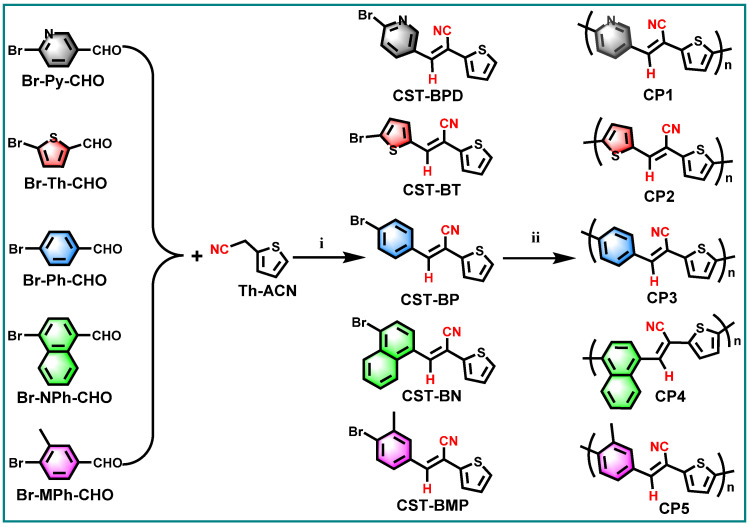
Two-step synthesis CST-based conjugated polymers **CP1**–**CP5**: (i) Knoevenagel condensation: *t*-BuONa/piperidine, MeOH/EtOH, RT/reflux; (ii) Pd(OAc)_2_/Pd_2_(dba)_3_, P(t-Bu)_2_Me-HBF_4_/P(o-MeOPh)_3_, K_2_CO_3_/Cs_2_CO_3_, PivOH, and toluene under Ar at 150/110 °C.

**Figure 2 molecules-28-02203-f002:**
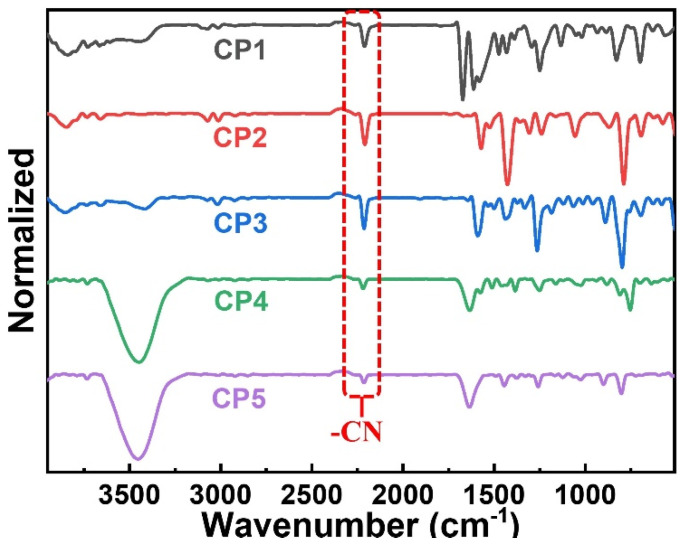
FT-IR spectra of **CP1**–**CP5**.

**Figure 3 molecules-28-02203-f003:**
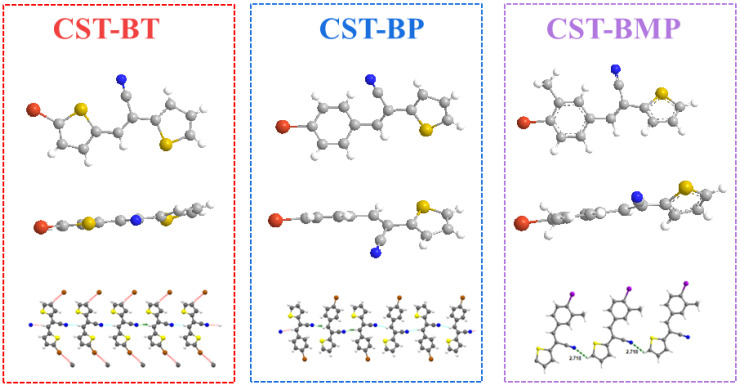
Single-crystal structure of the **CST-BT**, **CST-BP**, and **CST-BMP**.

**Figure 4 molecules-28-02203-f004:**
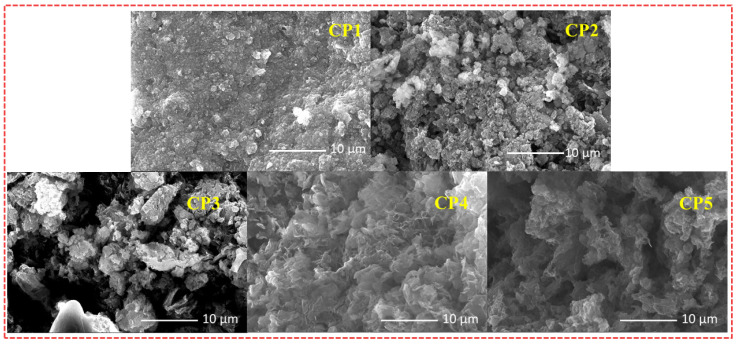
SEM images of **CP1**–**CP5**.

**Figure 5 molecules-28-02203-f005:**
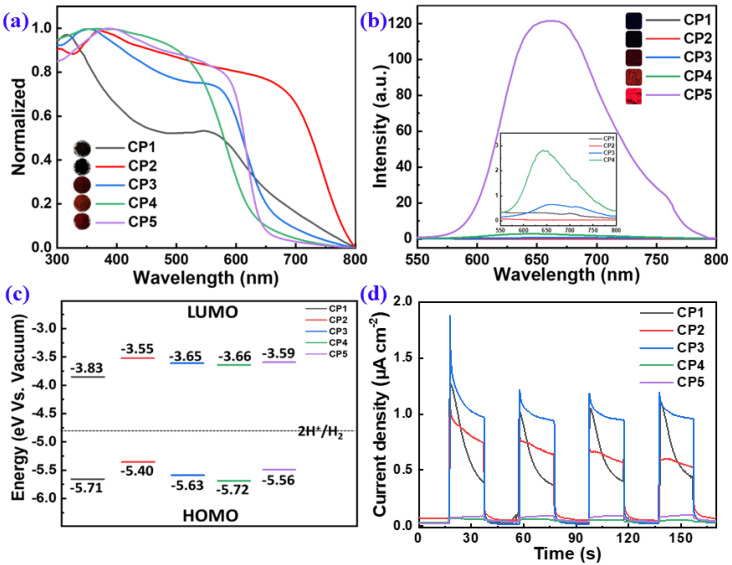
UV-vis DRS (**a**) and steady-state PL (**b**) spectra, FMO alignments (**c**), and TPR analysis (**d**) of **CP1**–**CP5**. [Insets in (**a**,**b**) show the photos of **CP1**–**CP5** under ambient conditions and 365 nm irradiation, respectively.].

**Figure 6 molecules-28-02203-f006:**
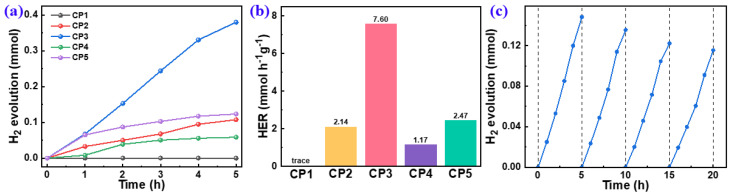
PHP as a function of time of 10 mg CPs dispersed in AA/H_2_O under visible light (**a**) and normalized HERs (**b**), based on a PHP cycle test of **CP3** (**c**).

## Data Availability

Not applicable.

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
