# Peer review of "Tunable Donor–Acceptor Linear Conjugated Polymers Involving Cyanostyrylthiophene Linkages for Visible-Light-Driven Hydrogen Production"

_molecules, 2023, doi:10.3390/molecules28052203_

Round 1
Reviewer 1 Report
Diarylethenes-based conjugated polymers (CP), such as MEH-PPV and PTV, as a kind of classical pi-functional materials, have been widely used as high charge mobility orgainic semiconductors for organic electronic devices, and developed as low-cost donor materials in organic solar cells very recently. In this work, Liu et al describe a one-component and atom- and step-economic DArP synthetic strategy to access cyanostyrylthiophenes (CST)-based linear D-A CPs for hydrogen production from photocatalytic water reduction. The structure-property-performance correlation of these CST-based CPs with varied building blocks are systematically studied, revealing that phenyl-CST-based CP3 exhibits a superior hydrogen evolution rate (HER) compared to most of the reported linear CPs. Systematical characterizations, including X-ray single crystal analysis, have been employed, among which CST-BT and CST-BMP are two newly reported crystals. Overall, this is an interesting work that will be instructive to the design of D-A CPs for photocatalytic applications. I would like to recommend it for publication in Molecules after minor revision.
1) CP3 afforded the highest HER among five CPs, I would like to suggest the authors to make a further comparison with other reported linear CPs.
2) This manuscript is related to cyanostyrylthiophene derivatives. Hence, for the statement “Monomers CST-BPD, CST-BT, CST-BP, CST-BN and CST-BMP were synthesized by the same procedure via Knoevenagel condensation” in page 7, more relevant references are suggested to be provided.
3) The quality of some figures can be improved, e.g., the SEM image in Figure 4. The size of Figures S4 and S5 are different.
4) Each figure, including the ones in Supporting Information should have been discussed in the main text, rather than leaving them unexplained.
Author Response
Response to Reviewer 1 Comments
Diarylethenes-based conjugated polymers (CP), such as MEH-PPV and PTV, as a kind of classical pi-functional materials, have been widely used as high charge mobility orgainic semiconductors for organic electronic devices, and developed as low-cost donor materials in organic solar cells very recently. In this work, Liu et al describe a one-component and atom- and step-economic DArP synthetic strategy to access cyanostyrylthiophenes (CST)-based linear D-A CPs for hydrogen production from photocatalytic water reduction. The structure-property-performance correlation of these CST-based CPs with varied building blocks are systematically studied, revealing that phenyl-CST-based CP3 exhibits a superior hydrogen evolution rate (HER) compared to most of the reported linear CPs. Systematical characterizations, including X-ray single crystal analysis, have been employed, among which CST-BT and CST-BMP are two newly reported crystals. Overall, this is an interesting work that will be instructive to the design of D-A CPs for photocatalytic applications. I would like to recommend it for publication in Molecules after minor revision.
We appreciate the positive comments made by the reviewer: “ …The structure-property-performance correlation of these CST-based CPs with varied building blocks are systematically studied…. . Systematical characterizations, including X-ray single crystal analysis….are two newly reported crystals. Overall, this is an interesting work that will be instructive to the design of D-A CPs for photocatalytic applications. I would like to recommend it for publication in Molecules after minor revision …”.
Point 1: CP3 afforded the highest HER among five CPs, I would like to suggest the authors to make a further comparison with other reported linear CPs.
Response 1: Thanks for nice suggestion! To the best our knowledge, HER of 7.60 mmol h-1 g-1 is above the average among the abovementioned types of organic photocatalysts, and outperforming most of the reported linear CPs, as mentioned in the Discussion of “2.3 Photocatalytic Hydrogen Production Performances of the CPs” (Lines 200-202). In this new revision, a table involes HERs of the reported linear CPs has been added in SI for comparison (Table S1).
Point 2: This manuscript is related to cyanostyrylthiophene derivatives. Hence, for the statement “Monomers CST-BPD, CST-BT, CST-BP, CST-BN and CST-BMP were synthesized by the same procedure via Knoevenagel condensation” in page 7, more relevant references are suggested to be provided.
Response 2: Thanks for comment! We have carefully reexamined the literature, and other three relevant references have been provided as Refs 33, 34, and 35 in this revised version.
Point 3: The quality of some figures can be improved, e.g., the SEM image in Figure 4. The size of Figures S4 and S5 are different.
Response 3: Thanks for reminding. We have carefully re-checked all the figures and improved the quality, e.g., Figure 4 has been enlarged and their sizes have also been unified.
Point 4: Each figure, including the ones in Supporting Information should have been discussed in the main text, rather than leaving them unexplained.
Response 4: Thanks for pointing out. In this revised version, each figure of our manuscript has been properly cited and discussed.
Reviewer 2 Report
This paper presents a series of conjugated polymers prepared by Knoevenagel condensation for use in photocatalytic hydrogen production. Overall I felt that the study design was not adequate and that the presentation and impact of the work are fairly low, and would not recommend this article for publication. Issues include:
- The manuscript contains many typos, awkward word choices, or sentences that were not comprehensible. I am concious of the fact that writing in English for non-native speakers is a barrier to publication, but the writing seriously impeded scientific understanding of the work.
- The photophysical properties of the materials were not fully described. - The quantum yields of fluorescence are not quantified, but are only discussed qualitatively in terms of relative brightness to each other, which is not useful. Quantum yields should be obtained. Only normalized UV-vis spectra were presented and I could not find any molar absorptivity values, so the reader cannot tell how well these materials absorb light.
- SEM images: morphology of the polymers will depend on how they were prepared (precipitated, spin-coated from good solvent, lyophilized, etc.) I am not sure there is any useful information here, though the authors use the SEMs of polymer powders extensively to interpret structure-property relationships.
- The paper compares the photocatalytic hydrogen production of these 5 polymers, but does not compare these polymers to what is already known in the literature. Are these PHP values good, poor, outstanding? The reader would not know from this work. Furthermore, the authors should compare the PHP production of their new materials with a known standard to illustrate that the method they used is comparable.
- The polymer characterization was also incomplete; I could not find data on the molecular weights or dispersity of the polymers the authors obtained, and so do not know what the authors made. This would make reproducing the results of this study impossible.
- The authors extensively interpret the dihedral angles calculated by DFT as an explanation for the photophysical performance of their materials, but - these materials are polymers! There will be many twists of different magnitude along all of the polymer chains, and so I am not sure if this discussion can be considered correct either.
Author Response
Response to Reviewer 2 Comments
This paper presents a series of conjugated polymers prepared by Knoevenagel condensation for use in photocatalytic hydrogen production. Overall I felt that the study design was not adequate and that the presentation and impact of the work are fairly low, and would not recommend this article for publication.
Thanks for comment! In this revised version, we have carefully re-checked all the sentences and improved the presentation.
Diarylethenes-based conjugated polymers (CP), such as MEH-PPV and PTV, as a kind of classical π-functional materials, have been widely used as high charge mobility orgainic semiconductors for organic electronic devices. The novelty of the current work lies in that a series of cyanostyrylthiophenes (CST)-based linear D-A CPs are accessed via one-component and atom-economic DArP synthetic strategy, and developed as photocatalysts toward hydrogen production from water reduction. The structure-property-performance correlation of these CST-based CPs with varied building blocks are systematically studied, revealing that phenyl-CST-based CP3 exhibits a superior hydrogen evolution rate compared to the reported linear CPs. Systematical characterizations, including X-ray single crystal analysis, have been employed, among which CST-BT and CST-BMP are two newly reported crystals.
Point 1: The manuscript contains many typos, awkward word choices, or sentences that were not comprehensible. I am concious of the fact that writing in English for non-native speakers is a barrier to publication, but the writing seriously impeded scientific understanding of the work.
Response 1: Thanks for nice suggestion. We have carefully re-checked the whole manuscript to improve to the language quality and exclude typos.
Point 2: The photophysical properties of the materials were not fully described. The quantum yields of fluorescence are not quantified, but are only discussed qualitatively in terms of relative brightness to each other, which is not useful. Quantum yields should be obtained. Only normalized UV-vis spectra were presented and I could not find any molar absorptivity values, so the reader cannot tell how well these materials absorb light.
Response 2: Thanks very much! In this new revision, the intensities of the steady-stated PL spectra of CP1-CP5 have been added to make a clearer comparison among five CPs (Figure. 5b). As heterogeneous catalysts, CP1-CP5 have no solubility in common organic solvents owing to the lack of alkyl side chains, the PL spectra of CP1-CP5 were collected from their solid states instead of solutions, the quantum yields can’t be precisely evaluated. In the area of photocatalysis, the UV-vis and PL of insoluble heterogeneous CP photocatalysts are typically obtained as normalized spectra. Some examples: ACS Catal. 2023, 13, 204, https://doi.org/10.1021/acscatal.2c04993; Chem. Eng. J. 2023, 141553, https://doi.org/10.1016/j.cej.2023.141553.
Point 3: SEM images: morphology of the polymers will depend on how they were prepared (precipitated, spin-coated from good solvent, lyophilized, etc.) I am not sure there is any useful information here, though the authors use the SEMs of polymer powders extensively to interpret relationships.
Response 3: Thanks for valuable comment. The heterogeueous photocatalysts CP1-CP5 are insoluble all polymeric powders with relatively stable nanoµ-scale morphologies once they are formed. Here, we give a detailed discussion on the SEM morphology mainly because that the monomers at molecular levels might exert an effect on the conformation and aggregate of polymeric chains, which can thus be reflected by SEM morphologies at nanoµ-scale, and affect the electron transfer and separation of the polymer semiconductors at nanoscale.
Point 4: The paper compares the photocatalytic hydrogen production of these 5 polymers, but does not compare these polymers to what is already known in the literature. Are these PHP values good, poor, outstanding? The reader would not know from this work. Furthermore, the authors should compare the PHP production of their new materials with a known standard to illustrate that the method they used is comparable.
Response 4: Thanks for nice suggestion! To the best our knowledge, HER of 7.60 mmol h-1 g-1 is above the average among the abovementioned types of organic photocatalysts, and outperforming most of the reported linear CPs. In this new revision, a table involes HERs of the reported linear CPs has been added in SI (Table S1) for comparison, see the revised Discussion part “2.3 Photocatalytic Hydrogen Production Performances of the CPs” (Lines 200-202).
Point 5: The polymer characterization was also incomplete; I could not find data on the molecular weights or dispersity of the polymers the authors obtained, and so do not know what the authors made. This would make reproducing the results of this study impossible.
Response 5: Thanks for the valuable comment! Like most of the reported CP-based heterogeneous photocatalysts, CP1-CP5 are also insoluble in common organic solvents owing to the lack of alkyl side chains. Thus, the molecular weights of CP1-CP5 are unavailable, because that the solubility of the polymeric samples in mobile phases, such as THF, chloroform and chlorobenzene, is a prerequisite for the molecular weight measurement via GPC. FT-IR, as a most commonly used characterization method for heterogeneous photocatalysts, has revealed that the vinylene linkers and –CN groups have been successfully incorporated into the skeletons of CP1-CP5.
Point 6: The authors extensively interpret the dihedral angles calculated by DFT as an explanation for the photophysical performance of their materials, but these materials are polymers! There will be many twists of different magnitude along all of the polymer chains, and so I am not sure if this discussion can be considered correct either.
Response 6: Thanks for valuable comment. To more precisely reflect the conformation of polymer chains, here, the dihedral angles of dimers (instead of the monomers) of CP1-CP5 are carefully calculated by DFT. Although it is not in 100% accuracy, the conformation of dimer can still be used to predicate the dominant conformation of polymer, which will provide valuable information regarding the molecular geometries and opto-electronic properties of polymers, as supported by some related literatures: Tuning the mechanical and electric properties of conjugated polymer semiconductors: side-chain design based on asymmetric benzodithiophene building blocks. Adv. Funct. Mater., 2022, 32, 2203527; Cyanostyrylthiophene-based ambipolar conjugated polymers: synthesis, properties, and analyses of backbone fluorination effect. Macromolecules, 2018, 51, 966−976; Effects of π-conjugated bridges on photovoltaic properties of donor-π-acceptor conjugated copolymers. Macromolecules, 2013, 46, 1208-1216.
Round 2
Reviewer 1 Report
I donot have any further comments. I recommend it for publication in its present form.
Reviewer 2 Report
see below